# Imitation Learning from Observations by Minimizing Inverse Dynamics Disagreement

**Chao Yang**[1*], **Xiaojian Ma**[12*], **Wenbing Huang**[1*],
**Fuchun Sun**[1], **Huaping Liu**[1], **Junzhou Huang**[3], **Chuang Gan**[4]
[1] Beijing National Research Center for Information Science and Technology (BNRist),
State Key Lab on Intelligent Technology and Systems,
Department of Computer Science and Technology, Tsinghua University
[2] Center for Vision, Cognition, Learning and Autonomy, Department of Computer Science, UCLA
[3] Tencent AI Lab, [4] MIT-IBM Watson AI Lab
yangchao18@mails.tsinghua.edu.cn, maxiaojian@ucla.edu
hwenbing@126.com, fcsun@tsinghua.edu.cn

## Abstract

This paper studies *Learning from Observations (LfO)* for imitation learning with access to state-only demonstrations. In contrast to *Learning from Demonstration (LfD)* that involves both action and state supervision, LfO is more practical in leveraging previously inapplicable resources (*e.g.* videos), yet more challenging due to the incomplete expert guidance. In this paper, we investigate LfO and its difference with LfD in both theoretical and practical perspectives. We first prove that the gap between LfD and LfO actually lies in the disagreement of inverse dynamics models between the imitator and the expert, if following the modeling approach of GAIL [15]. More importantly, the upper bound of this gap is revealed by a negative causal entropy which can be minimized in a model-free way. We term our method as *Inverse-Dynamics-Disagreement-Minimization* (IDDM) which enhances the conventional LfO method through further bridging the gap to LfD. Considerable empirical results on challenging benchmarks indicate that our method attains consistent improvements over other LfO counterparts.

## 1 Introduction

A crucial aspect of intelligent robots is their ability to perform a task of interest by imitating expert behaviors from raw sensory observations [5]. Towards this goal, GAIL [15] is one of the most successful imitation learning methods, which adversarially minimizes the discrepancy of the occupancy measure between the agent and the expert for policy optimization. However, along with many other methods [31, 3, 29, 30, 1, 23, 6, 9, 13, 2], GAIL adopts a heavily supervised training mechanism, which demands not only the expert's state (*e.g.* observable spatial locations), but also its accurate action (*e.g.* controllable motor commands) performed at each time step.

Whereas providing expert action indeed enriches the information and hence facilitates the imitation learning process, collecting them could be difficult and sometimes infeasible for some certain practical cases, particularly when we would like to learn skills from a large number of internet videos. Besides, imitation learning under action guidance is not biologically reasonable [39], as our human can imitate skills through adjusting the action to match the demonstrators' state, without knowing what exact action the demonstrator has performed. To address these concerns, several methods have been proposed [35, 39, 5, 21, 36], including the one named GAIfO [40] that extends the idea of GAIL to

---

[*]Denotes equal contributions. Corresponding author: Fuchun Sun.

the case with the absence of action guidance. Joining the previous denotations, this paper will define the original problem as *Learning from Demonstrations (LfD)*, and the new action agnostic setting as *Learning from Observations (LfO)*.

Undoubtedly, conducting LfO is *non-trivial*. For many tasks (*e.g.* robotic manipulation, locomotion and video-game playing), the reward function depends on both action and state. It remains challenging to determine the optimal action corresponding to the best reward purely from experts' state observations, since there could be multiple choices of action corresponding to the same sequence of state in a demonstration—when, for example, manipulating redundant-degree robotic hands, there exist countless force controls of joints giving rise to the same pose change. Yet, realizing LfO is still possible, especially if the expert and the agent share the same dynamics system (namely, the same robot). In this condition and what this paper has assumed, the correlation between action and state can be learned by the self-playing of the agent (see for example in [39]).

In this paper, we approach LfO by leveraging the concept of *inverse dynamics disagreement minimization*. As its name implies, inverse dynamics disagreement is defined as the discrepancy between the inverse dynamics models of the expert and the agent. Minimizing such disagreement becomes the task of inverse dynamics prediction, a well-known problem that has been studied in robotics [24]. Interestingly, as we will draw in this paper, the inverse dynamics disagreement is closely related to LfD and LfO. To be more specific, we prove that the inverse dynamics disagreement actually accounts for the optimization gap between LfD and naive LfO, if we model LfD by using GAIL [15] and consider naive LfO as GAIfO [40]. This result is crucial, not only for it tells the quantitative difference between LfD and naive LfO but also for it enables us to solve LfO more elegantly by minimizing the inverse dynamics disagreement as well.

To mitigate the issue of inverse dynamics disagreement, here we propose a model-free solution for the consideration of efficiency. In detail, we derive an upper bound of the gap, which turns out to be a negative entropy of the state-action occupancy measure. Under the assumption of deterministic system, such entropy contains a mutual information term that can be optimized with the popularly-used tool (*i.e.* MINE [4]). For convenience, we term our method as the *Inverse-Dynamics-Disagreement-Minimization* (IDDM) based LfO in what follows. To verify the effectiveness of our IDDM, we perform experimental comparisons on seven challenging control tasks, ranging from traditional control to locomotion [8]. The experimental results demonstrate that our proposed method attains consistent improvements over other LfO counterparts.

The rest of the paper is organized as follows. In Sec. 2, we will first review some necessary notations and preliminaries. Then our proposed method will be detailed in Sec. 3 with theoretical analysis and efficient implementation, and the discussions with existing LfD and LfO methods will be included in Sec. 4. Finally, experimental evaluations and ablation studies will be demonstrated in Sec. 5.

## 2   Preliminaries

**Notations.**   To model the action decision procedure in our context, we consider a standard Markov decision process (MDP) [37] as $(\mathcal{S}, \mathcal{A}, r, \mathcal{T}, \mu, \gamma)$, where $\mathcal{S}$ and $\mathcal{A}$ are the sets of feasible state and action, respectively; $r(s, a) : \mathcal{S} \times \mathcal{A} \to \mathbb{R}$ denotes the reward function on state $s$ and action $a$; $\mathcal{T}(s'|s, a) : \mathcal{S} \times \mathcal{A} \times \mathcal{S} \to [0, 1]$ characterizes the dynamics of the environment and defines the transition probability to next-step state $s'$ if the agent takes action $a$ at current state $s$; $\mu(s) : S \to [0, 1]$ is the distribution of initial state and $\gamma \in (0, 1)$ is the discount factor. A stationary policy $\pi(a|s) : \mathcal{S} \times \mathcal{A} \to [0, 1]$ defines the probability of choosing action $a$ at state $s$. A temporal sequence of state-action pairs $\{(s_0, a_0), (s_1, a_1), \cdots\}$ is called a trajectory denoted by $\zeta$.

**Occupancy measure.**   To characterize the statistical properties of an MDP, the concept of occupancy measure [28, 38, 15, 17] is proposed to describe the distribution of state and action under a given policy $\pi$. Below, we introduce its simplest form, *i.e.*, *State Occupancy Measure*.

**Definition 1** (State Occupancy Measure)**.**   *Given a stationary policy $\pi$, state occupancy measure $\rho_\pi(s) : \mathcal{S} \to \mathbb{R}$ denotes the discounted state appearance frequency under policy $\pi$*

$$\rho_\pi(s) = \sum_{t=0}^{\infty} \gamma^t P(s_t = s|\pi). \tag{1}$$

With the use of state occupancy measure, we can define other kinds of occupancy measures under different supports, including state-action occupancy measure, station transition occupancy measure, and joint occupancy measure. We list their definitions in Tab. 1 for reader's reference.

**Inverse dynamics model.** We present the *inverse dynamics model* [34, 33] in Definition 2, which infers the action inversely given state transition $(s, s')$.

Table 1: Different occupancy measures for MDP

|  | State-Action Occupancy Measure | State Transition Occupancy Measure | Joint Occupancy Measure |
|---|---|---|---|
| Denotation | $\rho_\pi(s, a)$ | $\rho_\pi(s, s')$ | $\rho_\pi(s, a, s')$ |
| Support | $\mathcal{S} \times \mathcal{A}$ | $\mathcal{S} \times \mathcal{S}$ | $\mathcal{S} \times \mathcal{A} \times \mathcal{S}$ |
| Definition | $\rho_\pi(s)\pi(a\|s)$ | $\int_\mathcal{A} \rho_\pi(s, \overline{a})\mathcal{T}(s'\|s, \overline{a})d\overline{a}$ | $\rho_\pi(s, a)\mathcal{T}(s'\|s, a)$ |

**Definition 2** (Inverse Dynamics Model)**.** *Let $\rho_\pi(a\|s, s')$ denotes the density function of the inverse dynamics model under the policy $\pi$, whose relation with $\mathcal{T}$ and $\pi$ can be shown as follows.*

$$\rho_\pi(a|s, s') := \frac{\mathcal{T}(s'|s, a)\pi(a|s)}{\int_\mathcal{A} \mathcal{T}(s'|s, \overline{a})\pi(\overline{a}|s)d\overline{a}}. \tag{2}$$

## 3 Methodology

In this section, we first introduce the concepts of LfD, naive LfO, and *inverse dynamics disagreement*. Then, we prove that the optimization gap between LfD and naive LfO actually leads to the inverse dynamics disagreement. As such, we enhance naive LfO by further minimizing the inverse dynamics disagreement. We also demonstrate that such disagreement can be bounded by an entropy term and can be minimized by a model-free method. Finally, we provide a practical implementation for our proposed method.

### 3.1 Inverse Dynamics Disagreement: the Gap between LfD and LfO

**LfD.** In Sec. 1, we have mentioned that GAIL and many other LfD methods [15, 18, 16, 27] exploit the discrepancy of the occupancy measure between the agent and expert as a reward for policy optimization. Without loss of generality, we will consider GAIL as the representative LfD framework and build our analysis on this description. This LfD framework requires to compute the discrepancy over the state-action occupancy measure, leading to

$$\min_\pi \mathbb{D}_{\mathrm{KL}}\left(\rho_\pi(s, a)||\rho_E(s, a)\right), \tag{3}$$

where $\rho_E(s, a)$ denotes the occupancy measure under the expert policy, and $\mathbb{D}_{\mathrm{KL}}(\cdot)$ computes the Kullback-Leibler (KL) divergence[2]. We have omitted the policy entropy term in GAIL, but our following derivations will find that the policy entropy term is naturally contained in the gap between LfD and LfO.

**Naive LfO.** In LfO, the expert action is absent, thus directly working on $\mathbb{D}_{\mathrm{KL}}(\rho_\pi(s, a)||\rho_E(s, a))$ is infeasible. An alternative objective could be minimizing the discrepancy on the state transition occupancy measure $\rho_\pi(s, s')$, as mentioned in GAIfO [40]. The objective function in (3) becomes

$$\min_\pi \mathbb{D}_{\mathrm{KL}}\left(\rho_\pi(s, s')||\rho_E(s, s')\right). \tag{4}$$

We will refer this as *naive LfO* in the following context. Compared to LfD, the key challenge in LfO comes from the absence of action information, which prevents it from applying typical action-involved imitation learning approaches like behavior cloning [31, 3, 11, 29, 30] or apprenticeship learning [23, 1, 38]. Actually, action information can be implicitly encoded in the state transition $(s, s')$. We have assumed the expert and the agent share the same dynamics system $\mathcal{T}(s'|s, a)$. It is thus possible for us to learn the action-state relation by exploring the difference between their inverse dynamics models.

We define the inverse dynamics disagreement between the expert and the agent as follows.

**Definition 3** (Inverse Dynamics Disagreement). *Given expert policy $\pi_E$ and agent policy $\pi$, the inverse dynamics disagreement is defined as the KL divergence between the inverse dynamics models of the expert and the agent.*

$$\text{Inverse Dynamics Disagreement} := \mathbb{D}_{\text{KL}}\left(\rho_\pi(a|s, s')||\rho_E(a|s, s')\right). \tag{5}$$

Given a state transition $(s, s')$, minimizing the inverse dynamics disagreement is learning an optimal policy to fit the expert/ground-truth action labels. This is a typical robotic task [24], and it can be solved by using a mixture method of combining machine learning model and control model.

Here, we contend another role of the inverse dynamics disagreement in the context of imitation learning. Joining the denotations in (3), (4) and Definition 3, we provide the following result.

**Theorem 1.** *If the agent and the expert share the same dynamics system, the relation between LfD, naive LfO, and inverse dynamics disagreement can be characterized as*

$$\mathbb{D}_{\text{KL}}\left(\rho_\pi(a|s, s')||\rho_E(a|s, s')\right) = \mathbb{D}_{\text{KL}}\left(\rho_\pi(s, a)||\rho_E(s, a)\right) - \mathbb{D}_{\text{KL}}\left(\rho_\pi(s, s')||\rho_E(s, s')\right). \tag{6}$$

Theorem 1 states that the inverse dynamics disagreement essentially captures the optimization gap between LfD and naive LfO. As (5) is non-negative by nature, optimizing the objective of LfD implies minimizing the objective of LfO but not vice versa. One interesting observation is that when the action corresponding to a given state transition is unique (or equivalently, the dynamics $\mathcal{T}(s'|s, a)$ is injective *w.r.t $a$*), the inverse dynamics is invariant to different conducted policies, hence the inverse dynamics disagreement between the expert and the agent reduces to zero. We summarize this by the following corollary.

**Corollary 1.** *If the dynamics $\mathcal{T}(s'|s, a)$ is injective* w.r.t $a$, *LfD is equivalent to naive LfO.*

$$\mathbb{D}_{\text{KL}}\left(\rho_\pi(s, a)||\rho_E(s, a)\right) = \mathbb{D}_{\text{KL}}\left(\rho_\pi(s, s')||\rho_E(s, s')\right). \tag{7}$$

However, since most of the real world tasks are performed in rather complex environments, (5) is usually not equal to zero and the gap between LfD and LfO should not be overlooked, which makes minimizing the inverse dynamics disagreement become unavoidable.

## 3.2 Bridging the Gap with Entropy Maximization

We have shown that the inverse dynamics disagreement amounts to the optimization gap between LfD and naive LfO. Therefore, the key to improving naive LfO mainly lies in *inverse dynamics disagreement minimization*. Nevertheless, accurately computing the disagreement is difficult, as it relies on the environment dynamics $\mathcal{T}$ and the expert policy (see (2)), both of which are assumed to be unknown. In this section, we try a smarter way and propose an upper bound for the gap, without the access of the dynamics model and expert guidance. This upper bound is tractable to be minimized if assuming the dynamics to be deterministic. We introduce the upper bound by the following theorem.

**Theorem 2.** *Let $\mathcal{H}_\pi(s, a)$ and $\mathcal{H}_E(s, a)$ denote the causal entropies over the state-action occupancy measure of the agent and expert, respectively. When $\mathbb{D}_{\text{KL}}\left[\rho_\pi(s, s')||\rho_E(s, s')\right]$ is minimized, we have*

$$\mathbb{D}_{\text{KL}}\left[\rho_\pi(a|s, s')||\rho_E(a|s, s')\right] \leqslant -\mathcal{H}_\pi(s, a) + \text{Const}. \tag{8}$$

Now we take a closer look at $\mathcal{H}_\pi(s, a)$. Following the definition in Tab. 1, the entropy of state-action occupancy measure can be decomposed as the sum of the policy entropy and the state entropy by

$$\mathcal{H}_\pi(s, a) = \mathbb{E}_{\rho_\pi(s, a)}\left[-\log \rho_\pi(s, a)\right] = \mathbb{E}_{\rho_\pi(s, a)}\left[-\log \pi(a|s)\right] + \mathbb{E}_{\rho_\pi(s)}\left[-\log \rho_\pi(s)\right]$$
$$= \mathcal{H}_\pi(a|s) + \mathcal{H}_\pi(s). \tag{9}$$

For the first term, the policy entropy $\mathcal{H}_\pi(a|s)$ can be estimated via sampling similar to previous studies [15]. For the second term, we leverage the mutual information (MI) between $s$ and $(s', a)$ to obtain an unbiased estimator of the entropy over the state occupancy measure, namely,

---

**Algorithm 1** Inverse-Dynamics-Disagreement-Minimization (IDDM)

---

**Input:** State-only expert demonstrations $\mathcal{D}_E = \{\zeta_i^E\}$ where $\zeta_i = \{s_0^E, s_1^E, ...\}$, policy $\pi_\theta$, discriminator $D_\phi$, MI estimator $\mathcal{I}$, entropy weights $\lambda_p, \lambda_s$, maximum iterations $M$.

   **for** 1 to $M$ **do**

      Sample agent rollouts $\mathcal{D}_A = \{\zeta^i\}$, $\zeta^i \sim \pi_\theta$ and update the MI estimator $\mathcal{I}$ with $\mathcal{D}_A$.

      Update the discriminator $D_\phi$ with the gradient

$$\hat{\mathbb{E}}_{\mathcal{D}_A} \left[ \nabla_\phi \log D_\phi(s, s') \right] + \hat{\mathbb{E}}_{\mathcal{D}_E} \left[ \nabla_\phi \log(1 - D_\phi(s, s')) \right].$$

      Update policy $\pi_\theta$ using the following gradient (can be integrated into methods like PPO [32])

$$\hat{\mathbb{E}}_{\mathcal{D}_A}[\nabla_\theta \log \pi_\theta(a|s) Q(s,a)] - \lambda_p \nabla_\theta \mathcal{H}_{\pi_\theta}(a|s) - \lambda_s \nabla_\theta \mathcal{I}_{\pi_\theta}(s; (s', a)),$$

$$\text{where } Q(\bar{s}, \bar{a}) = \hat{\mathbb{E}}_{\mathbb{D}_A} \left[ \log D_\phi(s, s') | s_0 = \bar{s}, a_0 = \bar{a} \right].$$

   **end for**

---

$$\mathcal{H}_\pi(s) = \mathcal{I}_\pi(s; (s', a)) + \underbrace{\mathcal{H}_\pi(s|s', a)}_{=0} = \mathcal{I}_\pi(s; (s', a)), \qquad (10)$$

where we have $\mathcal{H}_\pi(s|s', a) = 0$ as we have assumed $(s, a) \to s'$ is a deterministic function[3]. In our implementation, the MI $\mathcal{I}_\pi(s; (s', a))$ is computed via maximizing the lower bound of KL divergence between the product of marginals and the joint distribution following the formulation of [25]. Specifically, we adopt MINE [4, 14] which implements the score function with a neural network to achieve a low-variance MI estimator.

**Overall loss.** By combining the results in Theorem 1, Theorem 2, (9) and (10), we enhance naive LfO by further minimizing the upper bound of its gap to LfD. The eventual objective is

$$\mathcal{L}_\pi = \mathbb{D}_{\text{KL}}(\rho_\pi(s, s') || \rho_E(s, s')) - \lambda_p \mathcal{H}_\pi(a|s) - \lambda_s \mathcal{I}_\pi(s; (s', a)), \qquad (11)$$

where the first term is from naive LfO, and the last two terms are to minimize the gap between LfD and naive LfO. We also add trade-off weights $\lambda_p$ and $\lambda_s$ to the last two terms for more flexibility.

**Implementation.** As our above derivations can be generalized to JS-divergence (see Sec. A.3-4 in the supplementary material), we can utilize the GAN-like [25] method to minimize the first term in (11). In detail, we introduce a parameterized discriminator network $D_\phi$ and a policy network $\pi_\theta$ (serves as a generator) to realize the first term in (11). The term $\log D_\phi(s, s')$ could be interpreted as an immediate cost since we minimize its expectation over the current occupancy measure. A similar training method can also be found in GAIL [15], but it relies on state-action input instead. We defer the derivations for the gradients of the causal entropy $\nabla \mathcal{H}_\pi(a|s)$ and MI $\nabla \mathcal{I}_\pi(s; (s', a))$ with respect to the policy in Sec. A.5 of the supplementary material. Note that the objective (11) can be optimized by any policy gradient method, like A3C [22] or PPO [32], and we apply PPO in our experiments. The algorithm details are summarized in Alg. 1.

## 4 Related Work

### 4.1 Learning from Demonstrations

Modern dominant approaches on LfD mainly fall into two categories: **Behavior Cloning (BC)** [31, 3, 29, 30], which seeks the best policy that can minimize the action prediction error in demonstration, and **Inverse Reinforcement Learning (IRL)** [23, 1], which infers the reward used by expert to guide the agent policy learning procedure. A notable implementation of the latter is GAIL [15], which reformulates IRL as an occupancy measure matching problem [28], and utilizes the GAN [12] method

along with a forward RL to minimize the discrepancy of occupancy measures between imitator and demonstrator. There are also several follow-up works that attempt to enhance the effectiveness of discrepancy computation [19, 16, 10, 27], whereas all these methods require exact action guidance at each time step.

## 4.2 Learning from Observations

There have already been some researches on exploring LfO. These approaches exploit either a complex hand-crafted reward function or an inverse dynamics model that predicts the exact action given state transitions. Here is a summary to show how they are connected to our method.

**LfO with Hand-crafted Reward and Forward RL.** Recently, Peng et al. propose **DeepMimic**, a method that can imitate locomotion behaviors from motion clips without action labeling. They design a reward to encourage the agent to directly match the expert's physical proprieties, such as joint angles and velocities, and run a forward RL to learn the imitation policy. However, as the hand-crafted reward function does not take expert action (or implicitly state transition) into account, it is hard to be generalized to tasks whose reward depends on actions.

**Model-Based LfO.** BCO [39] is another LfO approach. The authors infer the exact action from state transition with a learned inverse dynamics model (2). The state demonstrations augmented with the predicted actions deliver common state-action pairs that enable imitation learning via BC [31]. At its heart, the inverse dynamics model is trained in parallel by collecting rollouts in the environment. However, as showed in (2), the inverse dynamics model depends on the current policy, underlying that an optimal inverse dynamics model would be infeasible to obtain before the optimal policy is learned. The performance of BCO would thus be not theoretically guaranteed.

**LfO with GAIL.** GAIfO [40, 41] is the closest work to our method. The authors follow the formulation of GAIL [15] but replace the state-action definition $(s, a)$ with state transition $(s, s')$, which gives the same objective in Eq. (4) if replacing KL with JS divergence. As we have discussed in Sec. 3.1, there is a gap between Eq. (4) and the objective of original LfD in Eq. (3), and this gap is induced by inverse dynamics disagreement. Unlike our method, the solution by GAIfO never minimizes the gap and is thereby no better than ours in principal.

# 5 Experiments

For the experiments below, we investigate the following questions:

1. Does inverse dynamics disagreement really account for the gap between LfD and LfO?
2. With state-only guidance, can our method achieve better performance than other counterparts that do not consider inverse dynamics disagreement minimization?
3. What are the key ingredients of our method that contribute to performance improvement?

To answer the first question, we conduct toy experiments with the *Gridworld* environment [37]. We test and contrast the performance of our method (refer to Eq. (11)) against GAIL (refer to Eq. (3)) and GAIfO (refer to Eq. (4)) on the tasks under different levels of inverse dynamics disagreement. Regarding the second question, we evaluate our method against several baselines on six physics-based control benchmarks [8], ranging from low-dimension control to challenging high-dimension continuous control. Finally, we explore the ablation analysis of two major components in our method (the policy entropy term and the MI term) to address the last question. Due to the space limit, we defer more detailed specifications of all the evaluated tasks into the supplementary material.

## 5.1 Understanding the Effect of Inverse Dynamics Disagreement

This collection of experiments is mainly to demonstrate how inverse dynamics disagreement influences the LfO approaches. We first observe that inverse dynamics disagreement will increase when the number of possible action choices grows. This is justified in Fig. 1a, and more details about the relation between inverse dynamics disagreement and the number of action choices are provided in Sec. B.2 of the supplementary material. Hence, we can utilize different action scales to reflect

different levels of inverse dynamics disagreement in our experiments. Controlling the action scale in *Gridworld* is straightforward. For example in Fig. 1b, agent block (in red) may take various kinds of actions (walk, jump or others) for moving to a neighbor position towards the target (in green), and we can specify different numbers of action choices.

We simulate the expert demonstrations by collecting the trajectories of the policy trained by PPO [32]. Then we conduct GAIL, GAIfO, and our method, and evaluate the pre-defined reward values for the policies they learn. It should be noted that all imitation learning methods have no access to the reward function during training. As we can see in Fig. 1c, the gap between GAIL and GAIfO is growing as the number of action choices (equivalently the level of inverse dynamics disagreement) increases, which is consistent with our conclusion in Theorem 1. We also find that the rewards of GAIL and GAIfO are the same when the number of action choice is 1 (*i.e.* the dynamics is injective), which follows the statement in Corollary 1. Our method lies between GAIL and GAIfO, indicating that the gap between GAIL and GAIfO can be somehow mitigated by explicitly minimizing inverse dynamics disagreement. Note that, GAIL also encounters performance drop when inverse dynamics disagreement becomes large. This is mainly because the imitation learning problem itself also becomes more difficult when the dynamics is complicated and beyond injective.

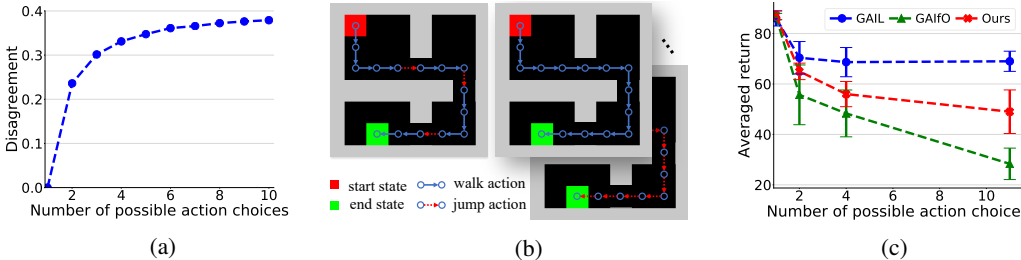

(a)           (b)           (c)

Figure 1: Toy examples on illustrating the effect of inverse dynamics disagreement.

## 5.2 Comparative Evaluations

For comparative evaluations, we carry out several LfO baselines, including DeepMimic [26], BCO [39], and GAIfO [40]. In particular, we introduce a modified version of GAIfO that only takes a single state as input to illustrate the necessity of leveraging state transition; we denote this method as GAIfO-s. We also run GAIL [15] to provide oracle reference. All experiments are evaluated within fixed steps. On each task, we run each algorithm over five times with different random seeds. In Fig. 2, the solid curves correspond to the mean returns, and the shaded regions represent the variance over the five runs. The eventual results are summarized in Tab. 2, which is averaged over 50 trials of the learned policies. Due to the space limit, we defer more details to the supplementary material.

Table 2: Summary of quantitative results. All results correspond to the original exact reward defined in [7]. *CartPole* is excluded from DeepMimic because no crafting reward is available.

|  | CartPole | Pendulum | DoublePendulum | Hopper | HalfCheetah | Ant |
|---|---|---|---|---|---|---|
| DeepMimic | - | 731.0±19.0 | 454.4±154.0 | 2292.6±1068.9 | 202.6±4.4 | -985.3±13.6 |
| BCO | 200.0±0.0 | 24.9±0.8 | 80.3±13.1 | 1266.2±1062.8 | 4557.2±90.0 | 562.5±384.1 |
| GAIfO | 197.5±7.3 | 980.2±3.0 | 4240.6±4525.6 | 1021.4±0.6 | 3955.1±22.1 | -1415.0±161.1 |
| GAIfO-s* | 200.0±0.0 | 952.1±23.0 | 1089.2±51.4 | 1022.5±0.40 | 2896.5±53.8 | -5062.3±56.9 |
| Ours | **200.0±0.0** | **1000.0±0.0** | **9359.7±0.2** | **3300.9±52.1** | **5699.3±51.8** | **2800.4±14.0** |
| GAIL | 200.0±0.0 | 1000.0±0.0 | 9174.8±1292.5 | 3249.9±34.0 | 6279.0±56.5 | 5508.8±791.5 |
| Expert | 200.0±0.0 | 1000.0±0.0 | 9318.8±8.5 | 3645.7±181.8 | 5988.7±61.8 | 5746.8±117.5 |

*GAIfO with single state only.

The results read that our method achieves comparable performances with the baselines on the easy tasks (such as *CartPole*) and outperforms them by a large margin on the difficult tasks (such as *Ant*, *Hopper*). We also find that our algorithm exhibits more stable behaviors. For example, the

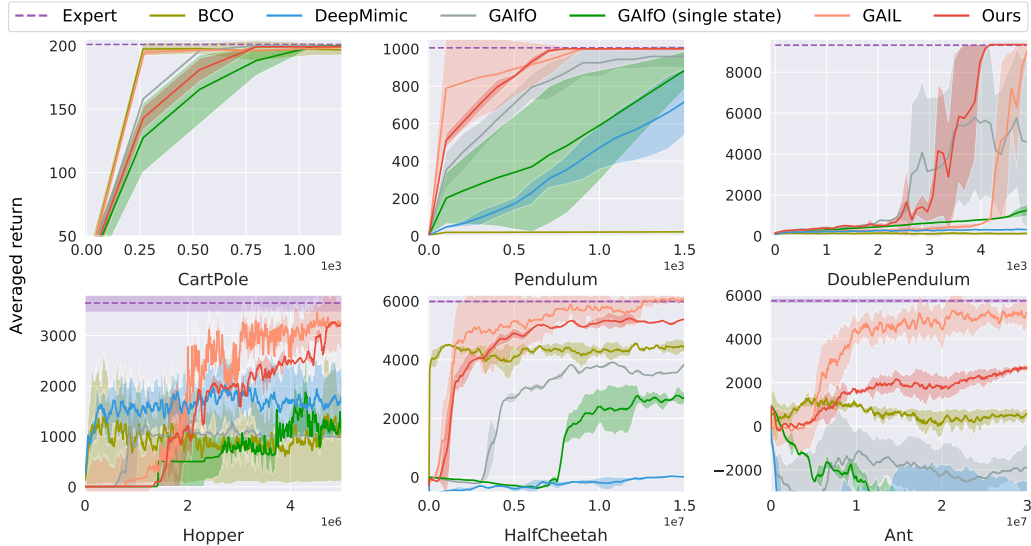

Figure 2: Learning curves under challenging robotic control benchmarks. For each experiment, a step represents one interaction with the environment. Detailed plots can be found in the supplementary.

performance of BCO on *Ant* and *Hopper* will unexpectedly drop down as the training continues. We conjecture that BCO explicitly but not accurately learns the inverse dynamics model from data, which yet is prone to over-fitting and leads to performance degradation. Conversely, our algorithm is model-free and guarantees the training stability as well as the eventual performance, even for the complex tasks including *HalfCheetah*, *Ant* and *Hopper*.

Besides, GAIfO performs better than GAIfO-s in most of the evaluated tasks. This illustrates the importance of taking state-transition into account to reflect action information in LfO. Compared with GAIfO, our method clearly attains consistent and significant improvements on *HalfCheetah* (+1744.2), *Ant* (+4215.0) and *Hopper* (+2279.5), thus convincingly verifying that minimizing the optimization gap induced by inverse dynamics disagreement plays an essential role in LfO, and our proposed approach can effectively bridge the gap. For the tasks that have relatively simple dynamics (*e.g. CartPole*), GAIfO achieves satisfying performances, which is consistent with our conclusion in Corollary 1.

DeepMimic that relies on hand-crafted reward struggles on most of the evaluated tasks. Our proposed method does not depend on any manually-designed reward signal, thus it becomes more self-contained and more practical in general applications.

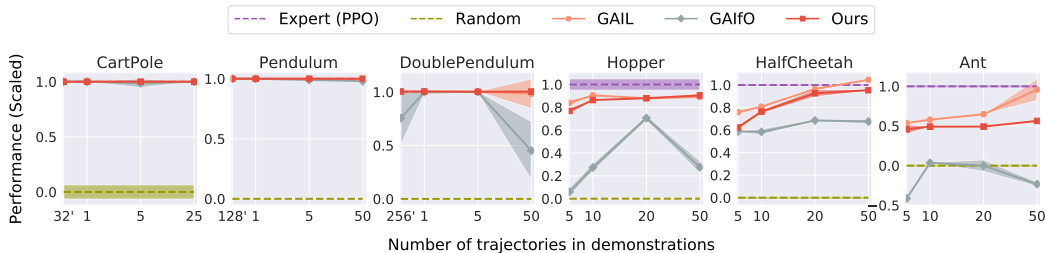

Figure 3: Comparative results of GAIL [15], GAIfO [40] and our method with different number of trajectories in demonstrations. The performance is the averaged cumulative return over 5 trajectories and has been scaled within [0, 1] (the random and the expert policies are fixed to be 0 and 1, respectively). We also conduct experiments with demonstrations containing state-action/state-transition pairs with the number less than that within one complete trajectory. We use $32'$, $128'$ and $256'$ pairs (denoted in the beginning of the x axes) for the first three tasks, respectively.

Finally, we compare the performances of GAIL, GAIfO and our method with different numbers of demonstrations. The results are presented in Fig. 3. It reads that for simple tasks like *CartPole* and *Pendulum*, there are no significant differences for all evaluated approaches, when the number of

demonstrations changes. While for the tasks with a higher dimension of state and action, our method performs advantageously over GAIfO. Even compared with GAIL that involves action demonstrations, our method still delivers comparable results. For all methods, more demonstrations facilitate better performances especially when the tasks become more complicated (*HalfCheetah* and *Ant*).

## 5.3   Ablation Study

The results presented in the previous section suggest that our proposed method can outperform other LfO approaches on several challenging tasks. Now we further perform a diverse set of analyses on assessing the impact of the policy entropy term and the MI term in  (11). As these two terms are controlled by $\lambda_p, \lambda_s$, we will explore the sensitivity of our algorithm in terms of their values.

**Sensitivity to Policy Entropy.**   We design four groups of parameters on *HalfCheetah*, where $\lambda_p$ is selected from $\{0, 0.0005, 0.001, 0.01\}$ and $\lambda_s$ is fixed at $0.01$. The final results are plotted in Fig. 4, with the learning curves and detailed quantitative results provided in the supplementary material. The results suggest that we can always promote the performances by adding policy entropy. Although different choices of $\lambda_p$ induce minor differences in their final performances, they are overall better than GAIfO that does not include the policy entropy term in its objective function.

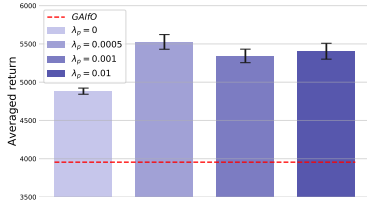

Figure 4: Sensitivity to the policy entropy weight $\lambda_p$.

**Sensitivity to Mutual Information.**   We conduct four groups of experiments on *HalfCheetah* by ranging $\lambda_s$ from 0.0 to 0.1 and fixing $\lambda_p$ to be 0.001. The final results are shown in Fig. 5 (the learning curves and averaged return are also reported in the supplementary material). It is observed that the imitation performances could always benefit from adding the MI term, and the improvements become more significant when the $\lambda_s$ has a relatively large magnitude. All of the variants of our method consistently outperform GAIfO, thus indicating the importance of the mutual information term in our optimization objective.

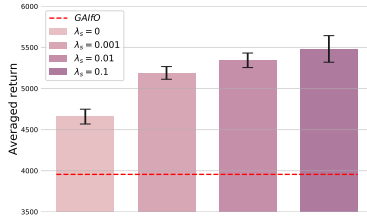

Figure 5: Sensitivity to the MI weight $\lambda_s$.

We also provide the results of performing a grid search on $\lambda_s$ and $\lambda_p$ in the supplementary material to further illustrate how better performance could be potentially obtained.

## 6   Conclusion

In this paper, our goal is to perform imitation Learning from Observations (LfO). Based on the theoretical analysis for the difference between LfO and Learning from Demonstrations (LfD), we introduce inverse dynamics disagreement and demonstrate it amounts to the gap between LfD and LfO. To minimize inverse dynamics disagreement in a principled and efficient way, we realize its upper bound as a particular negative causal entropy and optimize it via a model-free method. Our model, dubbed as *Inverse-Dynamics-Disagreement-Minimization* (IDDM), attains consistent improvement over other *LfO* counterparts on various challenging benchmarks. While our paper mainly focuses on control planning, further exploration on combining our work with representation learning to enable imitation across different domains could be a new direction for future work.

## Acknowledgments

This research was funded by National Science and Technology Major Project of the Ministry of Science and Technology of China (No.2018AAA0102900). It was also partially supported by National Science Foundation of China (Grant No.91848206), National Science Foundation of China (NSFC) and the German Research Foundation (DFG) in project Cross Modal Learning, NSFC 61621136008/DFG TRR-169. We would like to thank Mingxuan Jing and Dr. Boqing Gong for the insightful discussions and the anonymous reviewers for the constructive feedback.

## Footnotes

[2] The original GAIL method applies Jensen-Shannon (JS) divergence rather than KL divergence for measurement. Here, we will use KL divergence for the consistency throughout our derivations. Indeed, our method is also compatible with JS divergence, with the details provided in the supplementary material.

[3]In this paper, the tasks in our experiments indeed reveal deterministic dynamics. The mapping $(s, a) \to s'$ is deterministic also underlying that $(s', a) \to s$ is deterministic. Referring to [20], when $s, s', a$ are continuous, $\mathcal{H}(s|s', a)$ can be negative; but since these variables are actually quantified as finite-bit precision numbers (*e.g.* stored as 32-bit discrete numbers in computer), it is still true that conditional entropy is zero in practice.

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
