[Supplementary Material]

# Imitation Learning from Observations by Minimizing Inverse Dynamics Disagreement: Supplementary Material

**Chao Yang**[1*], **Xiaojian Ma**[12*], **Wenbing Huang**[1*],
**Fuchun Sun**[1], **Huaping Liu**[1], **Junzhou Huang**[3], **Chuang Gan**[4]
[1] Beijing National Research Center for Information Science and Technology (BNRist),
State Key Lab on Intelligent Technology and Systems,
Department of Computer Science and Technology, Tsinghua University
[2] Center for Vision, Cognition, Learning and Autonomy, Department of Computer Science, UCLA
[3] Tencent AI Lab
[4] MIT-IBM Watson AI Lab
yangchao18@mails.tsinghua.edu.cn, maxiaojian@ucla.edu
hwenbing@126.com, fcsun@tsinghua.edu.cn

## Contents

---

[*]Denotes equal contributions. Corresponding author: Fuchun Sun.

# A Proofs

## A.1 Theorem 1 and its Corollary

**Assumption 1.** *The environment dynamics is stationary between the expert and agent.*

**Lemma 1.** *The equality below holds.*

$$\mathbb{D}_{\mathrm{KL}}\left[\rho_\pi(s,a,s')||\rho_E(s,a,s')\right] = \mathbb{D}_{\mathrm{KL}}\left[\rho_\pi(s,a)||\rho_E(s,a)\right]. \tag{1}$$

*Proof.* We can expand the left side of the equality following Kullback–Leibler divergence definition and Assumption 1 as

$$\mathbb{D}_{\mathrm{KL}}\left[\rho_\pi(s,a,s')||\rho_E(s,a,s')\right]$$
$$= \mathbb{E}_{\rho_\pi}\left[\log\frac{\rho_\pi(s,a,s')}{\rho_E(s,a,s')}\right]$$
$$= \mathbb{E}_{\rho_\pi}\left[\log\frac{\rho_\pi(s,a)\mathcal{T}(s'|s,a)}{\rho_E(s,a)\mathcal{T}(s'|s,a)}\right]$$
$$= \mathbb{D}_{\mathrm{KL}}\left[\rho_\pi(s,a)||\rho_E(s,a)\right].$$

$\square$

**Theorem 1.** *The relation between LfD, naive LfO, and inverse dynamics disagreement can be characterized as*

$$\mathbb{D}_{\mathrm{KL}}\left(\rho_\pi(a|s,s')||\rho_E(a|s,s')\right) = \mathbb{D}_{\mathrm{KL}}\left(\rho_\pi(s,a)||\rho_E(s,a)\right) - \mathbb{D}_{\mathrm{KL}}\left(\rho_\pi(s,s')||\rho_E(s,s')\right). \tag{2}$$

*Proof.* We can subtract the Kullback-Leibler divergence between the state transition of expert and agent $\mathbb{D}_{\mathrm{KL}}(\rho_\pi(s,s')||\rho_E(s,s'))$ from the corresponding discrepancy over joint distribution $\mathbb{D}_{\mathrm{KL}}(\rho_\pi(s,a,s')||\rho_E(s,a,s'))$ as

$$\mathbb{D}_{\mathrm{KL}}\left(\rho_\pi(s,a,s')||\rho_E(s,a,s')\right) - \mathbb{D}_{\mathrm{KL}}\left(\rho_\pi(s,s')||\rho_E(s,s')\right)$$
$$= \int_{\mathcal{S}\times\mathcal{A}\times\mathcal{S}}\rho_\pi(s,a,s')\left(\log\frac{\rho_\pi(s,a,s')}{\rho_E(s,a,s')}\times\frac{\rho_E(s,s')}{\rho_\pi(s,s')}\right)dsdads'$$
$$= \int_{\mathcal{S}\times\mathcal{A}\times\mathcal{S}}\rho_\pi(s,a,s')\log\frac{\rho_\pi(a|s,s')}{\rho_E(a|s,s')}dsdads'$$
$$= \mathbb{D}_{\mathrm{KL}}\left(\rho_\pi(a|s,s')||\rho_E(a|s,s')\right). \tag{3}$$

With Lemma 1, we have

$$\mathbb{D}_{\mathrm{KL}}(\rho_\pi(s,a,s')||\rho_E(s,a,s')) - \mathbb{D}_{\mathrm{KL}}(\rho_\pi(s,s')||\rho_E(s,s'))$$
$$= \mathbb{D}_{\mathrm{KL}}(\rho_\pi(s,a)||\rho_E(s,a)) - \mathbb{D}_{\mathrm{KL}}(\rho_\pi(s,s')||\rho_E(s,s')). \tag{4}$$

With (3), (4)

$$\mathbb{D}_{\mathrm{KL}}\left(\rho_\pi(a|s,s')||\rho_E(a|s,s')\right) = \mathbb{D}_{\mathrm{KL}}\left(\rho_\pi(s,a)||\rho_E(s,a)\right) - \mathbb{D}_{\mathrm{KL}}\left(\rho_\pi(s,s')||\rho_E(s,s')\right).$$

$\square$

We now introduce the following lemma that will be used in the proof for Corollary 1.

**Lemma 2.** *if the dynamics is injective, i.e., distribution $\mathcal{T}(s'|s,a)$ that characterizes the dynamics is a degenerate distribution, and there is only one possible action corresponds to a state transition, the conditional distribution $\rho_\pi(a|s,s')$ that characterizes the corresponding inverse model under policy $\pi$ will also be injective, and will be independent with policy $\pi$, thus we have*

$$\rho_\pi(a|s,s') = \rho_E(a|s,s'), \tag{5}$$

*where $\rho_E(a|s,s')$ characterizes the corresponding inverse dynamics model for the expert.*

*Proof.* We will begin with the definition of inverse model as

$$\rho_\pi(a|s, s') = \frac{\mathcal{T}(s'|s, a)\pi(a|s)}{\int_A \mathcal{T}(s'|s, \bar{a})\pi(\bar{a}|s)d\bar{a}}. \tag{6}$$

Since $\mathcal{T}$ is injective, which means that $\mathcal{T}(s'|s, a) = \delta(s' - f(s, a)), f : \mathcal{S} \times \mathcal{A} \to \mathcal{S}, f$ is a deterministic function that independent with policy $\pi$, $\delta$ is Dirac delta function. When $f(s, a) = s'$, and for given $s, s'$, there is only one $a$ satisfy this equation, we have

$$\begin{aligned}\rho_\pi(a|s, s') &= \frac{\delta(0) \times \pi(a|s)}{1 \times \pi(\bar{a} = a|s)}\\ &= \delta(0).\end{aligned} \tag{7}$$

When $f(s, a) \neq s'$, it will be

$$\begin{aligned}\rho_\pi(a|s, s') &= \frac{0 \times \pi(a|s)}{\int_A \mathcal{T}(s'|s, \bar{a})\pi(a|s)d\bar{a}}\\ &= 0.\end{aligned} \tag{8}$$

Finally we can rewrite $\rho_\pi(a|s, s')$ as

$$\rho_\pi(a|s, s') = \begin{cases} \delta(0) & f(s, a) = s', \\ 0 & f(s, a) \neq s' \end{cases}, \tag{9}$$

which is independent with current policy $\pi$, thus we have

$$\rho_\pi(a|s, s') = \rho_E(a|s, s') = \begin{cases} \delta(0) & f(s, a) = s', \\ 0 & f(s, a) \neq s' \end{cases}. \tag{10}$$

$\square$

**Corollary 1.** *If the dynamics $\mathcal{T}(s'|s, a)$ is injective, LfD is equivalent to naive LfO.*

$$\mathbb{D}_{\mathrm{KL}}\left(\rho_\pi(s, a)||\rho_E(s, a)\right) = \mathbb{D}_{\mathrm{KL}}\left(\rho_\pi(s, s')||\rho_E(s, s')\right). \tag{11}$$

*Proof.* With Lemma 1, we can substitute the right side of the equality as

$$\begin{aligned}&\mathbb{D}_{\mathrm{KL}}(\rho_\pi(s, a)||\rho_E(s, a))\\ &= \mathbb{D}_{\mathrm{KL}}(\rho_\pi(s, a, s')||\rho_E(s, a, s'))\\ &= \mathbb{E}_{\rho_\pi}\left[\log \frac{\rho_\pi(s, a, s')}{\rho_\pi(s, a, s')}\right]\\ &= \mathbb{E}_{\rho_\pi}\left[\frac{\rho_\pi(s, s')\rho_\pi(a|s, s')}{\rho_E(s, s')\rho_E(a|s, s')}\right]\\ &= \underbrace{\mathbb{E}_{\rho_\pi}\left[\frac{\rho_\pi(s, s')}{\rho_E(s, s')}\right]}_{\text{by Lemma 2}}\\ &= \mathbb{D}_{\mathrm{KL}}(\rho_\pi(s, s')||\rho_E(s, s')).\end{aligned} \tag{12}$$

$\square$

## A.2 Theorem 2

**Theorem 2.** *Let $\mathcal{H}_\pi(s, a)$ and $\mathcal{H}_E(s, a)$ denote the causal entropies over the state-action occupancy measures of the agent and expert, respectively. When $\mathbb{D}_{\mathrm{KL}}\left[\rho_\pi(s, s')||\rho_E(s, s')\right]$ is minimized, we have*

$$\mathbb{D}_{\mathrm{KL}}\left[\rho_\pi(a|s, s')||\rho_E(a|s, s')\right] \leqslant -\mathcal{H}_\pi(s, a) + Const. \tag{13}$$

*Proof.* We will begin with the gap as the discrepancy between the inverse model of agent and expert

$$\mathbb{D}_{\mathrm{KL}}\left(\rho_\pi(a|s,s')||\rho_E(a|s,s')\right)$$

$$= \int_{\mathcal{S}\times\mathcal{A}\times\mathcal{S}} \rho_\pi(s,a,s') \log \frac{\rho_\pi(s,a,s')\rho_E(s,s')}{\rho_E(s,a,s')\rho_\pi(s,s')} dsdads'$$

$$= \underbrace{\int_{\mathcal{S}\times\mathcal{A}\times\mathcal{S}} \rho_\pi(s,a,s') \log \frac{\rho_\pi(s,a)\rho_E(s,s')}{\rho_E(s,a)\rho_\pi(s,s')} dsdads'}_{\text{by Lemma 1}}$$

$$= \underbrace{\int_{\mathcal{S}\times\mathcal{A}\times\mathcal{S}} \rho_\pi(s,a,s') \log \frac{\rho_\pi(s,a)}{\rho_E(s,a)} dsdads'}_{\mathbb{D}_{\mathrm{KL}}(\rho_\pi(s,s')||\rho_E(s,s'))=0}$$

$$= -\mathcal{H}_\pi(s,a) - \int_{\mathcal{S}\times\mathcal{A}} \rho_\pi(s,a) \log \rho_E(s,a) dsda$$

$$\leqslant -\mathcal{H}_\pi(s,a) + \sup_{\rho_\pi}\left(-\int_{\mathcal{S}\times\mathcal{A}} \rho_\pi(s,a) \log \rho_E(s,a) dsda\right)$$

$$= -\mathcal{H}_\pi(s,a) + Const. \tag{14}$$

Note that the second term in the inequality $\sup_{\rho_\pi}(\cdot)$ cannot be optimized *w.r.t.* the parameterized policy $\pi_\theta$ and thus can be omitted from the objective of maximizing $\mathcal{H}_\pi(s,a)$. $\square$

### A.3 Theorem 1 and its Corollary with Jensen-Shannon Divergence

**Lemma 3** (Lemma 1 with JS divergence). *The equality below holds.*

$$\mathbb{D}_{\mathrm{JS}}\left[\rho_\pi(s,a,s')||\rho_E(s,a,s')\right] = \mathbb{D}_{\mathrm{JS}}\left[\rho_\pi(s,a)||\rho_E(s,a)\right]. \tag{15}$$

*Proof.* We can expand the left side of the equality as

$$\mathbb{D}_{\mathrm{JS}}\left[\rho_\pi(s,a,s')||\rho_E(s,a,s')\right]$$

$$= \mathbb{E}_{\rho_\pi}\left[\frac{1}{2}\log\frac{\rho_\pi(s,a,s')}{\frac{1}{2}\rho_\pi(s,a,s')+\frac{1}{2}\rho_E(s,a,s')}\right] + \mathbb{E}_{\rho_E}\left[\frac{1}{2}\log\frac{\rho_E(s,a,s')}{\frac{1}{2}\rho_\pi(s,a,s')+\frac{1}{2}\rho_E(s,a,s')}\right]$$

$$= \mathbb{E}_{\rho_\pi}\left[\frac{1}{2}\log\frac{\rho_\pi(s,a)\mathcal{T}(s'|s,a)}{(\frac{1}{2}\rho_\pi(s,a)+\frac{1}{2}\rho_E(s,a))\mathcal{T}(s'|s,a)}\right]$$

$$\quad + \mathbb{E}_{\rho_E}\left[\frac{1}{2}\log\frac{\rho_E(s,a)\mathcal{T}(s'|s,a)}{(\frac{1}{2}\rho_\pi(s,a)+\frac{1}{2}\rho_E(s,a))\mathcal{T}(s'|s,a)}\right]$$

$$= \mathbb{E}_{\rho_\pi}\left[\frac{1}{2}\log\frac{\rho_\pi(s,a)}{\frac{1}{2}\rho_\pi(s,a)+\frac{1}{2}\rho_E(s,a)}\right] + \mathbb{E}_{\rho_E}\left[\frac{1}{2}\log\frac{\rho_E(s,a)}{\frac{1}{2}\rho_\pi(s,a)+\frac{1}{2}\rho_E(s,a)}\right]$$

$$= \mathbb{D}_{\mathrm{JS}}\left[\rho_\pi(s,a)||\rho_E(s,a)\right]. \tag{16}$$

$\square$

**Theorem 3** (Theorem 1 with JS divergence). *Given optimal expert policy $\pi_E$, $\rho_E(s,a)$, $\rho_E(s,s')$ denote its state-action and state transition occupancy measures accordingly, the optimization gap between minimizing the discrepancy of these two types of occupancy measures w.r.t. agent policy $\pi$ shows that*

$$\mathbb{D}_{\mathrm{JS}}\left(\rho_\pi(a|s,s')||\rho_E(a|s,s')\right) = \mathbb{D}_{\mathrm{JS}}\left(\rho_\pi(s,a)||\rho_E(s,a)\right) - \mathbb{D}_{\mathrm{JS}}\left(\rho_\pi(s,s')||\rho_E(s,s')\right) + \epsilon, \tag{17}$$

*where the minor term $\epsilon$ will converge to zero when minimizing the näive LfO objective under JS divergence $\mathbb{D}_{\mathrm{JS}}\left(\rho_\pi(s,s')||\rho_E(s,s')\right)$.*

*Proof.* We will begin with subtracting the Jensen-Shannon divergence between the state transition of expert and agent $\mathbb{D}_{\mathrm{JS}}(\rho_\pi(s,s')||\rho_E(s,s'))$ from the corresponding discrepancy over joint distribution $\mathbb{D}_{\mathrm{JS}}(\rho_\pi(s,a)||\rho_E(s,a))$ as

$$\mathbb{D}_{\mathrm{JS}}(\rho_\pi(s,a)||\rho_E(s,a)) - \mathbb{D}_{\mathrm{JS}}(\rho_\pi(s,s')||\rho_E(s,s'))$$

$$= \underbrace{\mathbb{D}_{\mathrm{JS}}(\rho_\pi(s,a,s')||\rho_E(s,a,s')) - \mathbb{D}_{\mathrm{JS}}(\rho_\pi(s,s')||\rho_E(s,s'))}_{\text{by Lemma 3}}$$

$$= \int_{\mathcal{S}\times\mathcal{A}\times\mathcal{S}} \frac{1}{2}\rho_\pi(s,a,s') \log\left(\frac{\rho_\pi(s,a,s')}{\frac{1}{2}(\rho_\pi(s,a,s')+\rho_E(s,a,s'))} \times \frac{\frac{1}{2}(\rho_\pi(s,s')+\rho_E(s,s'))}{\rho_\pi(s,s')}\right) dsdads'$$

$$+ \int_{\mathcal{S}\times\mathcal{A}\times\mathcal{S}} \frac{1}{2}\rho_E(s,a,s') \log\left(\frac{\rho_E(s,a,s')}{\frac{1}{2}(\rho_\pi(s,a,s')+\rho_E(s,a,s'))} \times \frac{\frac{1}{2}(\rho_\pi(s,s')+\rho_E(s,s'))}{\rho_E(s,s')}\right) dsdads'$$

$$= \int_{\mathcal{S}\times\mathcal{A}\times\mathcal{S}} \Big( \frac{1}{2}\rho_\pi(s,a,s')\left(\log\rho_\pi(a|s,s') + \log f(s,a,s')\right)$$

$$+ \frac{1}{2}\rho_E(s,a,s')\left(\log\rho_E(a|s,s') + \log f(s,a,s')\right)\Big) dsdads'. \tag{18}$$

We denotes (18) as $\Gamma(s,a,s')$ and $f(s,a,s') = \frac{\rho_\pi(s,s')+\rho_E(s,s')}{\rho_\pi(s,a,s')+\rho_E(s,a,s')}$. We then expand the discrepancy between the inverse model of expert and agent as

$$\mathbb{D}_{\mathrm{JS}}(\rho_\pi(a|s,s')||\rho_E(a|s,s'))$$

$$= \int_{\mathcal{S}\times\mathcal{A}\times\mathcal{S}} \frac{1}{2}\rho_\pi(s,a,s') \log\frac{2\rho_\pi(a|s,s')}{\rho_\pi(a|s,s')+\rho_E(a|s,s')} dsdads'$$

$$+ \int_{\mathcal{S}\times\mathcal{A}\times\mathcal{S}} \frac{1}{2}\rho_E(s,a,s') \log\frac{2\rho_E(a|s,s')}{\rho_\pi(a|s,s')+\rho_E(a|s,s')} dsdads'.$$

When $\mathbb{D}_{\mathrm{JS}}(\rho_\pi(s,s')||\rho_E(s,s')) \to 0$, there will be $\frac{\rho_\pi(s,s')}{\rho_E(s,s')} \to 1$, and $g(s,a,s') \to 2$. Therefore $\Gamma(s,a,s') - \mathbb{D}_{\mathrm{JS}}(\rho_\pi(a|s,s')||\rho_E(a|s,s')) \to 0$. If we denote

$$\epsilon = \Gamma(s,a,s') - \mathbb{D}_{\mathrm{JS}}(\rho_\pi(a|s,s')||\rho_E(a|s,s'))$$

$$= \mathbb{D}_{\mathrm{JS}}(\rho_\pi(s,a)||\rho_E(s,a)) - \mathbb{D}_{\mathrm{JS}}(\rho_\pi(s,s')||\rho_E(s,s')) - \mathbb{D}_{\mathrm{JS}}(\rho_\pi(a|s,s')||\rho_E(a|s,s')),$$

we get $\epsilon \to 0$ during the minimization of $\mathbb{D}_{\mathrm{JS}}(\rho_\pi(s,s')||\rho_E(s,s'))$. $\qquad\square$

**Corollary 2** (Corollary 1 with JS divergence). *If the dynamics $\mathcal{T}(s'|s,a)$ is injective, LfD is equivalent to naive LfO (replacing KL with JS divergence).*

$$\mathbb{D}_{\mathrm{JS}}\left(\rho_\pi(s,a)||\rho_E(s,a)\right) = \mathbb{D}_{\mathrm{JS}}\left(\rho_\pi(s,s')||\rho_E(s,s')\right). \tag{19}$$

*Proof.* With Lemma 3, we can substitute the right side of the equality as

$$\mathbb{D}_{\mathrm{JS}}(\rho_\pi(s,a)||\rho_E(s,a))$$

$$= \mathbb{D}_{\mathrm{JS}}(\rho_\pi(s,a,s')||\rho_E(s,a,s'))$$

$$= \mathbb{E}_{\rho_\pi}\left[\frac{1}{2}\log\frac{2\rho_\pi(s,a,s')}{\rho_\pi(s,a,s')+\rho_E(s,a,s')}\right] + \mathbb{E}_{\rho_E}\left[\frac{1}{2}\log\frac{2\rho_E(s,a,s')}{\rho_\pi(s,a,s')+\rho_E(s,a,s')}\right]$$

$$= \mathbb{E}_{\rho_\pi}\left[\frac{1}{2}\log\frac{2\rho_\pi(s,s')\rho_\pi(a|s,s')}{\rho_\pi(s,s')\rho_\pi(a|s,s')+\rho_E(s,s')\rho_E(a|s,s')}\right]$$

$$+ \mathbb{E}_{\rho_E}\left[\frac{1}{2}\log\frac{2\rho_E(s,s')\rho_E(a|s,s')}{\rho_\pi(s,s')\rho_\pi(a|s,s')+\rho_E(s,s')\rho_E(a|s,s')}\right]$$

$$= \underbrace{\mathbb{E}_{\rho_\pi}\left[\frac{1}{2}\log\frac{2\rho_\pi(s,s')}{\rho_\pi(s,s')+\rho_E(s,s')}\right] + \mathbb{E}_{\rho_E}\left[\frac{1}{2}\log\frac{2\rho_E(s,s')}{\rho_\pi(s,s')+\rho_E(s,s')}\right]}_{\text{by Lemma 2}}$$

$$= \mathbb{D}_{\mathrm{JS}}(\rho_\pi(s,s')||\rho_E(s,s')). \tag{20}$$

$$\square$$

## A.4 Theorem 2 with Jensen-Shannon Divergence

**Assumption 2.** *Given the inverse dynamics model of agent $\rho_\pi(a|s,s')$ and expert $\rho_E(a|s,s')$. The following inequality*

$$\mathbb{D}_{\mathrm{KL}}(\rho_E(a|s,s')||\rho_\pi(a|s,s')) \leqslant \mathbb{D}_{\mathrm{KL}}(\rho_\pi(a|s,s')||\rho_E(a|s,s')) + \delta, \tag{21}$$

*where $\delta$ is a minor term that will converge to 0 and thus can be omitted during the minimization the inverse dynamics disagreement between $\rho_\pi(a|s,s')$ and $\rho_E(a|s,s')$, should always holds, or the reverse Kullback-Leibler divergence of the inverse model between agent and expert should be bounded by the KL divergence between them.*

Note that, this assumption is somewhat trivial since when KL divergence $\mathbb{D}_{\mathrm{KL}}(\rho_\pi(a|s,s')||\rho_E(a|s,s'))$ is sufficiently minimized, the total variance between $\rho_\pi(a|s,s')$ and $\rho_E(a|s,s')$ is also minimized, thus inverse $\mathbb{D}_{\mathrm{KL}}(\rho_E(a|s,s')||\rho_\pi(a|s,s'))$ will be minimized at the same time. Apparently, the inequality holds and there will be $\delta \to 0$.

**Theorem 4** (Theorem 2 with JS divergence). *Let $\mathcal{H}_\pi(s,a)$ and $\mathcal{H}_E(s,a)$ denote the causal entropies over the state-action occupancy measures of the agent and expert, respectively. When $\mathbb{D}_{\mathrm{KL}}[\rho_\pi(s,s')||\rho_E(s,s')]$ is minimized, we have*

$$\mathbb{D}_{\mathrm{JS}}[\rho_\pi(a|s,s')||\rho_E(a|s,s')] \leqslant -\mathcal{H}_\pi(s,a) + \textit{Const}. \tag{22}$$

*Proof.* We will begin with the gap as the discrepancy between the inverse model of agent and expert

$$\mathbb{D}_{\mathrm{JS}}[\rho_\pi(a|s,s')||\rho_E(a|s,s')]$$

$$= \int_{\mathcal{S}\times\mathcal{A}\times\mathcal{S}} \frac{1}{2}\rho_\pi(s,a,s') \log \frac{2}{1 + \frac{\rho_E(a|s,s')}{\rho_\pi(a|s,s')}} ds da ds'$$

$$+ \int_{\mathcal{S}\times\mathcal{A}\times\mathcal{S}} \frac{1}{2}\rho_E(s,a,s') \log \frac{2}{1 + \frac{\rho_\pi(a|s,s')}{\rho_E(a|s,s')}} ds da ds'$$

$$= \log 2 - \int_{\mathcal{S}\times\mathcal{A}\times\mathcal{S}} \frac{1}{2}\rho_\pi(s,a,s') \log \left( 1 + \frac{\rho_E(a|s,s')}{\rho_\pi(a|s,s')} \right) ds da ds'$$

$$- \int_{\mathcal{S}\times\mathcal{A}\times\mathcal{S}} \frac{1}{2}\rho_E(s,a,s') \log \left( 1 + \frac{\rho_\pi(a|s,s')}{\rho_E(a|s,s')} \right) ds da ds'$$

$$\leqslant \log 2 + \int_{\mathcal{S}\times\mathcal{A}\times\mathcal{S}} \frac{1}{2}\rho_\pi(s,a,s') \log \frac{\rho_\pi(a|s,s')}{\rho_E(a|s,s')} ds da ds'$$

$$+ \int_{\mathcal{S}\times\mathcal{A}\times\mathcal{S}} \frac{1}{2}\rho_E(s,a,s') \log \frac{\rho_E(a|s,s')}{\rho_\pi(a|s,s')} ds da ds'$$

$$= \frac{1}{2}\mathbb{D}_{\mathrm{KL}}(\rho_\pi(a|s,s')||\rho_E(a|s,s')) + \frac{1}{2}\mathbb{D}_{\mathrm{KL}}(\rho_E(a|s,s')||\rho_\pi(a|s,s')) + \log 2$$

$$= \underbrace{\mathbb{D}_{\mathrm{KL}}(\rho_\pi(a|s,s')||\rho_E(a|s,s')) + \delta}_{\text{by Assumption 2}} + \log 2$$

$$\leqslant \underbrace{-\mathcal{H}_\pi(s,a) + \textit{Const}}_{\text{by Theorem 2}}, \tag{23}$$

where the first inequality is given by Jensen's inequality, and as we demonstrated in Assumption 2, the minor error $\delta$ will converge to 0 and thus can be omitted from the objective of minimizing $\mathbb{D}_{\mathrm{JS}}(\rho_\pi(a|s,s')||\rho_E(a|s,s'))$. $\square$

## A.5 Gradient Estimation of Policy Entropy and Mutual Information

Here we provide the policy gradient formula for causal entropy $\mathcal{H}_\pi(a|s)$ and mutual information $\mathcal{I}(s;(s',a))$.

**Proposition 1.** *The policy gradient of causal entropy $\mathcal{H}_\pi(a|s)$ is given by*

$$\nabla_\theta \mathbb{E}_{\pi_\theta}[-\log \pi_\theta(a|s)] = \mathbb{E}_\theta \left[ \nabla_\theta \log \pi_\theta(a|s) Q^H(s,a) \right],$$
$$\textit{where } Q^H(\bar{s},\bar{a}) = \mathbb{E}_{\pi_\theta}[-\log \pi_\theta(a|s)|s_0 = \bar{s}, a_0 = \bar{a}]. \tag{24}$$

*Proof.* Define $\rho(s) = \sum_a \rho(s, a)$ as the state occupancy measure. Then we have

$$
\begin{aligned}
\nabla_\theta \mathbb{E}_{\pi_\theta} \left[ -\log \pi_\theta(a|s) \right] &= -\nabla_\theta \sum_{s,a} \rho_{\pi_\theta}(s, a) \log \pi_\theta(a|s) \\
&= -\sum_{s,a} (\nabla_\theta \rho_{\pi_\theta}(s, a)) \log \pi_\theta(a|s) - \sum_s \rho_{\pi_\theta}(s) \sum_a \pi_\theta(a|s) \nabla_\theta \log \pi_\theta(a|s) \\
&= -\sum_{s,a} (\nabla_\theta \rho_{\pi_\theta}(s, a)) \log \pi_\theta(a|s) - \underbrace{\sum_s \rho_{\pi_\theta}(s) \sum_a \nabla_\theta \pi_\theta(a|s)}_{=0} \\
&= -\sum_{s,a} (\nabla_\theta \rho_{\pi_\theta}(s, a)) \log \pi_\theta(a|s).
\end{aligned}
\tag{25}
$$

This is exactly the policy gradient for RL with fixed cost function $c(s, a) = \log \pi_\theta(a|s)$. And the resulting policy gradient (24) is given by the standard policy gradient with cost $c(s, a)$. $\qquad\square$

**Proposition 2.** *The policy gradient of mutual information $\mathcal{I}(s; (s', a))$ is given by*

$$
\begin{aligned}
&\nabla_\theta \mathcal{I}_{\pi_\theta}(s; (s', a)) = \mathbb{E}_\theta \left[ \nabla_\theta \log \pi_\theta(a|s) Q^I(s, a) \right], \\
&\text{where } Q^I(\bar{s}, \bar{a}) = \mathcal{I}_{\pi_\theta} \left( s; (s', a) | s_0 = \bar{s}, a_0 = \bar{a} \right).
\end{aligned}
\tag{26}
$$

*Proof.* Note that $\mathcal{I}_\pi(s; (s', a)) = \mathcal{H}_\pi(s) - \mathcal{H}_\pi(s|s', a)$. The same as the proof for Proposition 1, we have

$$
\begin{aligned}
\nabla_\theta \mathcal{I}_{\pi_\theta}(s; (s', a)) &= -\nabla_\theta \sum_s \rho_{\pi_\theta}(s) \log \pi_\theta(s) - \nabla_\theta \sum_{s,a,s'} \rho_{\pi_\theta}(s, a, s') \log \pi_\theta(s|s', a) \\
&= -\sum_s (\nabla_\theta \rho_{\pi_\theta}(s)) \log \pi_\theta(s) - \sum_{s,a,s'} (\nabla_\theta \rho_{\pi_\theta}(s, a, s')) \log \pi_\theta(s|s', a).
\end{aligned}
\tag{27}
$$

We can thus see $\mathcal{I}_\pi(s; (s', a))$ as a fixed cumulative cost sum of a MDP, thus the resulting policy gradient will be (26). $\qquad\square$

# B  Specifications

## B.1  Hyperparameters

Tab. 1 lists the parameters for BCO [8], DeepMimic [5], GAIL [3], GAIfO [9] and proposed method used in the comparative evaluation.

Table 1: Hyperparameters for Evaluated Algorithms

| Parameter | Value |
|---|---|
| *Shared* | |
|     Optimizer | Adam [4] |
|     Learning rate | $3e^{-4}$ |
|     Batch size | 512 |
|     Discount ($\gamma$) | 0.99 |
|     Architecture of policy, value and discriminator networks | (300, 400) |
|     Nonlinearity | Tanh |
| *BCO* | |
|     Inverse model training epochs | 50 |
| *DeepMimic* | |
|     Reward type | Joint angle $q_t$ and Joint velocity $\dot{q}_t$ |
|     Reward design | $r_t = w_p \exp(-2(\sum_j \|\hat{q}_t^j - q_t^j\|_2))$ $\quad + w_v \exp(-0.1(\sum_j \|\hat{\dot{q}}_t^j - \dot{q}_t^j\|_2))$ |
|     Reward weight ($w_p, w_v$) | (0.8, 0.2) |
| *GAIL* | |
|     Weight of policy entropy | 0.01 |
| *Ours* | |
|     Weight of policy entropy ($\lambda_p$) | 0.01 |
|     Weight of state entropy ($\lambda_s$) | 0.1 |
|     Pretrained MI estimator steps | 10000 |
|     Update MI estimator steps | 50 |
|     Architecture of MI estimator network | (512, 512) |

## B.2  Gridworld Environment and Inverse Dynamics Disagreement

We will first demonstrate how our Gridworld environments are motivated by illustrating the relation between inverse dynamics disagreement and possible functional-equivalent action choices. Then we will provide the detailed specifications of our Gridworld environment.

The intuition behind the design of these experiments is that the complexity of the dynamics shows positive correlation with inverse dynamics disagreement. Under the deterministic dynamics, the complexity will be mainly dominated by the numbers of action choices (or size of state space, but we override it as we adopt a fixed size maze). Consider a MDP with two state $s_0$, $s_1$ and a set of actions $\{a_{01}^{(0)}, a_{01}^{(1)}, a_{01}^{(2)}, \cdots\}$ that can let the agent transform from $s_0$ to $s_1$. To approximately compute inverse dynamics disagreement, we denote $\pi(a|s = s_0) \sim \text{Categorical}(p_1 = p_2 = \cdots = p_k)$ is a uniformed initialized policy on a discrete action space with size $k$, $\pi_\theta(a|s = s_0)$ is a $\theta$-parameterized expert policy. Without loss of generality, we assume $\theta$ has a prior of normal distribution $\theta \sim \mathcal{N}(\mathbf{0}^k, \mathbf{1}^k)$. Therefore, we can approximately compute inverse dynamics disagreement as follows.

$$\begin{aligned}
&\textit{Inverse Dynamics Disagreement} \\
&\approx \mathbb{E}_{\theta \sim \mathcal{N}(\mathbf{0}^k, \mathbf{1}^k)} \left[ \mathbb{D}_{\text{KL}} \left( \pi(a|s = s0)p(s = s0) || \pi_\theta(a|s = s0)p(s = s0) \right) \right],
\end{aligned} \tag{28}$$

where $p(s = s_0)$ is the distribution of state. Since there is only two states available, $p(s = s_0) = 1$. Fig. 1a in the main paper are plotted with (28). As we can see, inverse dynamics disagreement does show a growing trend as the number of possible action choices increases.

To this end, we design several simple Gridworld environments (see Fig. 1) to help understand how inverse dynamics disagreement affects the imitation learning algorithms. The red block is the starting point of the agent, while the agent is encouraged to move toward the target green block. All the black and dark grey block are permitted to move through, while the grey block represents wall. The action that the agent may conduct including four basic ones: *moving left*, *moving right*, *moving up*, *moving down* when the number of possible action choices is one. If the number of possible action choices is larger than one (*e.g.* $n$ choices), there will be $n-1$ functional equivalent choices added to each original moving action, *i.e.*, now there will be $n$ action choices for moving left/right/up/down. For the reward strategy, once the agent successfully reaches the green target block, it will receive a reward of $100$, and the game will immediately come to an end. When the agent takes an original moving action, it will receive a penalty of $-1$, but when the agent chooses an action choice that is other than the original one, it will receive a penalty of $-5$. All the numerical evaluation results are under this strategy.

Figure 1: Gridworld environment.

## B.3 Other Environments

Tab. 2 lists the specifications about the benchmark environments and number of state transition pairs (state-action pairs for GAIL) in demonstration for each environment.

Table 2: Specifications for Evaluated Environments

| Environment | $\mathcal{S}$ | $\mathcal{A}$ | Max-Step | Demonstration Size |
|---|---|---|---|---|
| CartPole | $\mathbb{R}^4$ | $\{0, 1\}$ | 200 | 5000 |
| Pendulum | $\mathbb{R}^4$ | $\mathbb{R}^1$ | 1000 | 50000 |
| DoublePendulum | $\mathbb{R}^{11}$ | $\mathbb{R}^1$ | 1000 | 50000 |
| Hopper | $\mathbb{R}^{11}$ | $\mathbb{R}^3$ | 1000 | 50000 |
| Halfcheetah | $\mathbb{R}^{17}$ | $\mathbb{R}^6$ | 1000 | 50000 |
| Ant | $\mathbb{R}^{111}$ | $\mathbb{R}^8$ | 1000 | 50000 |

## C  Additional Empirical Results

### C.1  Quantitative Results of Toy Example

Tab. 3 lists the quantitative results of the toy Gridworld experiments.

Table 3: Quantitative Results of GAIL, GAIfO and our method in Gridworld Environment.

| Num. of Action | 1 | 2 | 4 | 11 |
|---|---|---|---|---|
| GAIL [3] | 86.0±3.0 | 70.4±6.4 | 68.7±5.8 | 69±4.0 |
| GAIfO [9] | 86.8±1.3 | 55.7±11.9 | 48.3±9.3 | 28.3±6.2 |
| Ours | 87.3±1.8 | 65.0±3.3 | 56.0±5.0 | 49.0±8.6 |

## C.2 Comparative Evaluations

**On the differences on results compared with [9]** For the baseline results of GAIfO [9], we notice that there are some differences between the results reported in [9] and our paper (Tab. 2 and Fig. 2 in the main paper). We hypothesis that the reason is twofold. First, **different physics engine**. Referring to the footnote 2 in page 5 of [10], the experiments in [9] are conducted with PyBullet [2] physics engine, while we use MuJoCo [7] instead since it is the default physics engine in OpenAI Gym [1] benchmark. Second, **different expert demonstrations**. As [9] does not provide the expert demonstrations used for imitation learning, we collect the demonstrations for all the baselines and our method by training an expert with PPO [6], which may lead to different imitation learning results.

## C.3 Quantitative Results of Ablation Study

Tab. 4 and Tab. 5 list the quantitative results of the ablations analysis (sensitivity to policy entropy and mutual information), while the corresponding learning curves can be found in Fig. 2a and Fig. 2b respectively.

Table 4: Quantitative results about $\lambda_p$ on *HalfCheetah* task.

| hyperparameters | Averaged return |
|---|---|
| $\lambda_p = 0.0, \lambda_s = 0.01$ | $4882.8 \pm 40.1$ |
| $\lambda_p = 0.0005, \lambda_s = 0.01$ | $5526.2 \pm 95.6$ |
| $\lambda_p = 0.001, \lambda_s = 0.01$ | $5343.2 \pm 88.5$ |
| $\lambda_p = 0.01, \lambda_s = 0.01$ | $5404.8 \pm 103.7$ |

Table 5: Quantitative results about $\lambda_s$ on *HalfCheetah* task.

| hyperparameters | Averaged return |
|---|---|
| $\lambda_p = 0.001, \lambda_s = 0.0$ | $4658.0 \pm 90.2$ |
| $\lambda_p = 0.001, \lambda_s = 0.001$ | $5189.7 \pm 77.2$ |
| $\lambda_p = 0.001, \lambda_s = 0.01$ | $5343.2 \pm 88.5$ |
| $\lambda_p = 0.001, \lambda_s = 0.1$ | $5540.5 \pm 100.3$ |

To further illustrate how our method can benefit from the two components (policy entropy and MI terms), here we also provide the results of performing a grid search on $\lambda_p$ and $\lambda_s$ in Fig. 3. All the numerical results are evaluated under the same criteria as other experiments.

The results read that, adding MI term can always promote the imitation performances, and the improvement can be more significant as the value of $\lambda_s$ increases. And the promotions it obtains are robust to the changes of $\lambda_p$. On the other hand, imitation performance can also benefit from adding policy entropy, while different $\lambda_p$ may lead to different improvements over the GAIfO baseline (the left-bottom block, with $\lambda_s = \lambda_p = 0$).

## References

[1] Greg Brockman, Vicki Cheung, Ludwig Pettersson, Jonas Schneider, John Schulman, Jie Tang, and Wojciech Zaremba. Openai gym, 2016.

|                    |                    |
|:------------------:|:------------------:|
| (a)                | (b)                |

Figure 2: **(a)** Learning curves of our method under different $\lambda_p$ settings. **(b)** Learning curves of our method under different $\lambda_s$ settings.

Figure 3: Quantitative results of a grid search on $\lambda_s$ and $\lambda_p$.

[2] Erwin Coumans and Yunfei Bai. Pybullet, a python module for physics simulation in robotics, games and machine learning. 2016-2017.

[3] Jonathan Ho and Stefano Ermon. Generative adversarial imitation learning. In *Advances in Neural Information Processing Systems (NeurIPS)*, 2016.

[4] Diederik P Kingma and Jimmy Ba. Adam: A method for stochastic optimization. In *International conference on Learning Representation (ICLR)*, 2015.

[5] Xue Bin Peng, Pieter Abbeel, Sergey Levine, and Michiel van de Panne. Deepmimic: Example-guided deep reinforcement learning of physics-based character skills. *ACM Transactions on Graphics (TOG)*, 2018.

[6] John Schulman, Filip Wolski, Prafulla Dhariwal, Alec Radford, and Oleg Klimov. Proximal policy optimization algorithms. *arXiv preprint arXiv:1707.06347*, 2017.

[7] Emanuel Todorov, Tom Erez, and Yuval Tassa. Mujoco: A physics engine for model-based control. In *IEEE/RSJ International Conference on Intelligent Robots and Systems (IROS)*, 2012.

[8] Faraz Torabi, Garrett Warnell, and Peter Stone. Behavioral cloning from observation. In *International Joint Conference on Artificial Intelligence (IJCAI)*, 2018.

[9] Faraz Torabi, Garrett Warnell, and Peter Stone. Generative adversarial imitation from observation. *arXiv preprint arXiv:1807.06158*, 2018.

[10] Faraz Torabi, Garrett Warnell, and Peter Stone. Imitation learning from video by leveraging proprioception. In *International Joint Conference on Artificial Intelligence (IJCAI)*, 2019.