[Reviews · NeurIPS 2019]

Reviewer 1



*** Update after reading review: Thank you for your feedback. I was happy with your inclusion of experiments on manipulation tasks, and agree they're convincing. I was also happy with your explanation on GAILfo vs GAIL vs your algorithm, and your discussion on Sun et al 2019. Your decision to release code also helps with any fears I have about reproducibility. I have changed my score to an 8 to reflect these improvements. ** Original Review Well written paper. Easy to follow the logic and thoughts of the authors. Fairly interesting line of thought. I feel like I learned something by reading it. It’s certainly of interest to the IRL community at this conference. I’m not sure the contributions would be of high interest outside the IRL community (for example, to the wider planning, RL, or inverse optimal control communities). But, that’s probably not so much of an issue. Without the code, it is difficult for me to evaluate their GAIL baseline, which can be difficult to get correct. This gives me pause, because in my personal experience with these algorithms GAIL does not usually care much if it receives the actions or not. Not a large demerit as I’m assuming the code will be released. But it does make things difficult to fully evaluate in their present state. “Provably Efficient Imitation Learning from Observation Alone” Should probably be cited. I found myself wanting to skip over the experiments in section 5.1. I understand the point that was being made but the discussion just couldn’t hold my interest, even though I read it several times. I think the environment feels contrived. I did check the math behind theorem 1 in the appendix. From a quick glance, everything seemed reasonable. Figure 2 indicates that the introduced algorithm does better than BC and GAIL on a variety of locomotion tasks. I do realize these benchmarks are fairly standard, but are they really the best choice of environments here? It seems like learning from observations would be more useful on robotic manipulation tasks, where it’s often easier to provide demonstrations via VR or even human demonstration, but less easy to collect actions. In MuJoCo environments, you always know the ground truth actions anyways. So pretending you don’t have them comes off as somewhat artificial. I don’t see it in the paper, but it seems like the locomotion baselines are provided via something like PPO? Although this is of course standard, it always seems funny to me when learning from demonstration papers use RL to provide expert demonstrations. It’s been shown that RL is a lot easier to optimize against than actual human demonstrations or even solutions provided via model based methods. I do appreciate that the author’s inherited this problem directly from GAIL and related literature, but it’s just hard for me to get excited about IRL results that don’t have a human in the loop, or at least some harder manipulation environments. These locomotion benchmarks are an odd choice for imitation learning. I think when we’re evaluating this work, we have to ask ourselves two key questions: 1. Does this work provide a significant improvement or insight over GAIL. 2. Does this work provide a significant improvement or insight over “Generative Adversarial Imitation from Observations” and “Imitation Learning from Video by Leveraging Proprioception” As for 1: Table 2 suggests that in practice the algorithm gets similar results to GAIL. The author’s do suggest inclusion of actions is important for GAIL’s performance but I am not so sure. I would need to see a much more detailed analysis of GAIL’s performance with and without actions to really make a decision about this. I think the analysis is interesting and the ideas are novel, so the paper can get by on that. However, I do have serious concerns that in practice the improvement over GAIL is rather small and no one will use this algorithm as an actual replacement for GAIL. Since, again, most people that I know that use GAIL already exclude actions. The correction terms to account for that DO seem to provide GAIN, but with these locomotion tasks being a narrow slice of what’s out there, it’s hard for me to feel completely confident. As for 2: I do think the existence of prior art considering this problem is a demerit for this paper. However, neither of those references considers the relationship between inverse dynamics models and GAIL. So, I think the paper is okay on this dimension.

Reviewer 2



This work shows that the difference between the state-action occupancy of LFD and the state-transition occupancy of LFO can be characterized by the difference between the inverse dynamics models of the expert policy and the agent's policy. This paper then proposes an algorithm for learning from observations by minimizing an upper bound on the inverse-dynamics disagreement. The method is evaluated on a number of benchmark tasks. I think the paper is generally well written and presents the derivation of IDDM in a fairly clear manner. The overall concept of inverse dynamics disagreement is also quite interesting. There are some minor typos and instances of awkward phrasing, but can be fixed with some additional polishing. The experiments show promising results on a diverse set of tasks. IDDM compares pretty favourable to a number of previous methods, though the improvements appear to be fairly marginal on many of the tasks. But overall, I am in favour of accepting this work. I have only some small concerns regarding some of the experiments, but it should in no way be a barrier for acceptance. In the performance statistics reported in Table 2, the performance of the handcrafted reward function (DeepMimic) seems unusually low, particularly for the ant. In the original paper, the method was able to reproduce very challenging skills with much more complex simulated agents. The tasks in this work are substantially simpler, so one would expect that the hand designed reward function, with some tuning, should be able to closely reproduce the demonstrations.

Reviewer 3



The authors study Learning from Observation (LfO) inverse reinforcement learning (imitation based on states alone, as opposed to states and actions as is done in LfD). They attempt to build on the GAIfO algorithm to acheive increased performance. To do so, they derive an expression for the performance gap between LfO and LfD. They then derive an upper bound on this expression that they add to the loss function to help close this optimization gap. They report significantly improved perfomance on a wide range of tasks over against a wide range of alternative approaches, including GAIfO. Notes: Eq 5 The explanation here is a little confusing. Firstly, the equation implies the divergence of LfD is always greater than that of LfO, since the inverse dynamics disagreement is always non-negative. This seems counter intuitive and could do with more explanation. Is the idea supposed to be that the KL of divergence could be zero, but the problem would still not be completely solved? If so, this bears clarification. Additionally the sentence "Eq 5 is not equal to zero by nature" is misleading, since there are cases when the IDD is zero (such as when \pi = E). Infact, one such case is highlighted in Corrolary 1. This should be restated with more specificity and clarity. Eq 10 Does this mean that this approach only works for deterministic state transitions? Or does the approach still more or less work, but with an additional source of error. Given the rarity of deterministic transition functions in real-world systems, it seems like it should be much more clearly stated that you are restricting your scope to deterministic systems. If the approach can still be employed effectively on non-deterministic systems without significant performance loss, then that should be stated and defended more clearly. Learning Curves: No comparison for 'number of interactions with the environment' vs GAIL. This seems conspicuously absent. It isn't strictly necessary, since the method has already shown to be superior to the others in its category, but it's odd that it's not here. No learning curves for 'number of demonstrations' Since the main motivation for LfO over LfD is 'availability of data', it seems odd that the authors only report learning curves for 'interactions with the environment' and not 'number of demonstrations'. It would especially be interesting to know at what point, if any, IDDM with more demonstrations begins to outperform GAIL with few demonstrations. GAIfO reported this, I see no reason it shouldn't be reported here as well. Originality: Derives a novel bound on accuracy and optimizes it, achieving state of the art results. The proof, bound, and algorithm are all novel and interesting. Although they build on an existing method and propose only a small change to the learning objective, the change they propose has significant impact and is very well motivated. Quality: The theoretical work is great, the proofs seem fine. The experimental work is not quite as well documented. Although it may not seem this way at first glance, their method somewhat increases the complexity by adding another neural network approximate the mutual information for the loss function, making their approach possibly more unweildy than GAIfO. Coupled with the fact that code was not included, these results may be difficult to replicate. Experiments are generally presented well and show good results, although there are a few experiments/plots that should have been included that were not. Clarity: For the most part, the paper is clearly written and very easy to follow. The supplemental material is also very easy to follow and well above the expected quality of writing. However, it should have been pointed out much more clearly that, as written, the approach presented only works in deterministic environments. Significance: The bound they prove is interesting and useful. Not only do they highlight a fundamental gap in this problem, they also show an effective way to contend with that gap and open up the possibility of new approaches to closing it in the future.

[Author Response · NeurIPS 2019]

We thank all reviewers for their constructive comments and are glad that our contributions are largely recognized.
Below, we address the reviewer's concerns point by point.

**To Reviewer #2:**

**Q1. Additional experiments on harder environments:** We agree with the reviewer that experiments of robotic
manipulation tasks other than locomotion benchmarks will further emphasize the benefits of our method. Hence
in Tab. A, we provide results of three MuJoCo manipulation examples: *Pusher*, *Striker* and *Thrower*. It reads that our
method achieves significant improvement over GAIfO and behaves comparably to GAIL. Applying our method for real
human demonstrations is a direct extension of our paper, but it demands more practical considerations (*e.g.* domain
adaption from human to robot, feature extraction of observations, etc) and will be left for future exploration.

**Q2. Our improvement or insight over GAIL:** Naive exclusion of actions from GAIL leads to GAIfO, which is
demonstrated to perform much worse than GAIL from Table 1 in the paper. In contrast, by bridging the gap between
GAIL and GAIfO, our method is able to outperform all other LfO baselines. We believe that our method is preferably a
practical choice for imitation learning from observations when GAIL is no longer applicable.

**Q3. Codes and reference citation:** We thank the reviewer for the reminding. Our code, including detailed instructions
on reproducing the results will be made public. Besides, we will cite the reference [Sun et al., 2019] raised by the
reviewer. In spite of focusing on the same topic, [Sun et al., 2019] provides theoretical guarantee on the sampling
efficiency of LfO over pure RL, while our core insight is improving LfO by investigating the gap between LfO and LfD.

**To Reviewer #3:**

**Q1. Results of DeepMimic:** Per the reviewer's suggestion, we have tuned the reward function of DeepMimic on
HalfCheetah and Ant by adjusting the weights of different reward terms. Even after careful tuning, DeepMimic is still
much worse than our method as observed from Tab. B. Moreover, applying DeepMimic requires to design the hand-craft
reward case by case, which makes it impracticable or even inapplicable for diverse types of agent mechanisms.

**Q2. On additional tasks:** The task with discrete actions has already been included in the main paper (namely,
*GridWorld* and *CartPole*), on which our method still performs promisingly. Due to the time limit and the major focus of
this work, we would like to make image-based imitation like Atari a future research as the reviewer suggested.

Table A: Additional experiments on manipulation tasks.

|  | Expert | GAIL | GAIfO | Ours |
|---|---|---|---|---|
| Pusher | -21.0±2.1 | -20.2±1.0 | -31.1±6.9 | -21.8±1.3 |
| Striker | -101.5±35.9 | -118.3±6.0 | -178.4±13.9 | -127.6±8.3 |
| Thrower | -26.8±0.3 | -28.6±1.1 | -74.4±5.6 | -29.9±1.1 |

Table B: Reward tuning for DeepMimic.

|  | HalfCheetah | Ant |
|---|---|---|
| DeepMimic (Tuned) | 202.6±4.4 | -985.3±13.6 |
| Ours | 5699.3±51.8 | 1970.3±110.1 |

**To Reviewer #4:**

**Q1. The explanations of Eq. 5 :** We are sorry for the confusion caused by the explanations of Eq. 5. We will provide
more illustrations on the relationship between LfD and LfO including that the divergence of LfD is always greater than
LfO and optimizing LfD implies optimizing LfO but not vice versa. For the statement on Eq. $5 \neq 0$, we apologize for
the improper presentation and will add necessary restrictions to make it consistent with Corollary 1.

**Q2. The clarification to deterministic systems:** We thank the reviewer for reminding this and will make the applicable
scope of our method (deterministic systems) clearer in the final revision. We also agree that employment to stochastic
dynamics is important for some real-world tasks and will be an exciting direction for future research.

**Q3. Learning curves for GAIL over number of interactions and over number of demos:** We provide the learning
curves under varying numbers of interactions for GAIL along with other methods in the left sub-figure of Fig. A.
Besides, the learning curves of GAIL, GAIfO, and our method with different numbers of demos on HalfCheetah are
reported in the right sub-figure of Fig. A (those on all other tasks will be included in the final revision due to the space
limit here). As indicated by the results, our method is able to outperform GAIL if the number of demos we use is
sufficiently larger than that of GAIL (*e.g.* our method with 50 demos vs. GAIL with 10 demos).

**Q4. The codes and replication of results:** Please refer to our response to Q3, Reviewer #2.

Figure A: **Left**: Learning curves w/ GAIL. **Right**: Results w/ different num. of demos on HalfCheetah task.

[Meta-Review · NeurIPS 2019]

Learning from Observation (LoF) is harder, but more practical, than Learning from Demonstration (LfD) that involves both action and state supervisions. The paper studies the difference between the two types of learning in both theoretical and practical perspectives, and relates the gap between LfD and LfO to inverse dynamics disagreement between the imitator and the expert. The paper includes an elaborate and interesting theoretical analysis of this gap, and proposes a method for bridging the gap through entropy maximization. The empirical evaluation is also thorough and includes both a toy problem for studying the effect of inverse dynamics discrepancy, MuJoCO problems and an ablation study. The reviewers are in agreement that this is a good, technically sound paper. The main issue raised by the reviewers is the fact that the proposed technique does not seem to improve significantly over GAIL, but the authors addressed this problem well in the rebuttal.